# MicroRNA Expression in Subretinal Fluid in Eyes Affected by Rhegmatogenous Retinal Detachment

**DOI:** 10.3390/ijms24033032

**Published:** 2023-02-03

**Authors:** Paolo Carpineto, Ester Sara Di Filippo, Agbeanda Aharrh Gnama, Danilo Bondi, Carla Iafigliola, Arturo Maria Licata, Stefania Fulle

**Affiliations:** 1Department of Medical, Oral and Biotechnological Sciences, University “G. d’Annunzio” of Chieti-Pescara, 66100 Chieti, Italy; 2Ophthalmology Clinic, University “G. d’Annunzio” of Chieti-Pescara, 66100 Chieti, Italy; 3Department of Neuroscience, Imaging and Clinical Sciences, University “G. d’Annunzio” of Chieti-Pescara, 66100 Chieti, Italy

**Keywords:** RRD, SRF, miRNAs, proliferative vitreoretinopathy, PVR, myopia

## Abstract

Proliferative vitreoretinopathy (PVR) is an abnormal intraocular scarring process that can complicate cases of rhegmatogenous retinal detachment (RRD). Although previous studies have examined the relevance of microRNAs (miRNAs) in ophthalmic diseases, only a few studies have evaluated the expression profiles of microRNAs in subretinal fluid. We hypothesized that the expression profiles of specific miRNAs may change in response to RRD, in the subretinal fluid that is directly in contact with photoreceptors and the retinal pigment epithelium (RPE). We looked for a potential correlation between the expression of specific miRNAs in eyes with RRD and known clinical risk factors of PVR. A total of 24 patients (59 ± 11 years) who underwent scleral buckling procedure were enrolled in this prospective study. Twenty-four undiluted subretinal fluid samples were collected, RNA was isolated and qRT-PCR was performed to analyze the expression of 12 miRNAs. We found the existence of a positive association between the expression of miR-21 (*p* = 0.017, r = 0.515) and miR-34 (*p* = 0.030, r = 0.624) and the duration of symptoms related to retinal detachment. Moreover, the expression of miR-146a tended to decrease in patients who developed PVR. Subretinal fluid constitutes an intriguing biological matrix to evaluate the role of miRNAs leading to the development of PVR.

## 1. Introduction

Proliferative vitreoretinopathy (PVR) indicates an abnormal intraocular scarring process that complicates up to 10% of cases of rhegmatogenous retinal detachment (RRD) [1], characterized by the formation of epiretinal and/or subretinal fibrotic cell membranes and intraretinal fibrosis, capable of contracting, generating a tractional retinal detachment or a mixed retinal detachment in case of the formation of new retinal breaks. The incidence of PVR has remained substantially unchanged in prospective studies despite the evolution of vitreoretinal surgical techniques over the past 25 years [2]. PVR is a multifactorial process, triggered by retinal detachment itself [3].

Retinal pigment epithelial (RPE) cells’ transdifferentiation plays a critical role in several retinal diseases, including hereditary retinal dystrophies and age-related macular degeneration, and it represents the key event in the pathogenesis of PVR [4]. Epithelial–Mesenchymal Transition (EMT) can be activated in pathological circumstances, such as inflammation, wound healing and carcinogenesis, allowing epithelial cells to achieve greater migration capacity and to increase their production of extracellular matrix components [5].

In addition to genetic and risk factors, non-coding RNAs are prominent in the pathophysiology of eye diseases and have been explored extensively. Within the retina, microRNAs (miRNAs) play central roles and there are promising possibilities to use them as disease biomarkers and master regulators for therapeutic strategies to stabilize or reset the cellular states of neurons and glial cells [6].

In the eye, various microRNAs are thought to play important roles in neuroprotection and angiogenesis [7]. Although previous studies have examined the relevance of microRNAs in ophthalmic diseases [8], only a few have evaluated the expression profiles of microRNAs in subretinal fluid (SRF) in eyes affected by RRD [9,10]. We hypothesized that the expression profiles of specific microRNAs may change in response to RRD, in the SRF that is directly in contact with photoreceptors and the retinal pigment epithelium (EPR), i.e., the main sites responsible for the visual prognosis after RRD and the pathogenesis of PVR. Therefore, we looked for a potential association between the expression of specific miRNAs in eyes with RRD and known clinical risk factors of PVR. For this purpose, 12 different specific miRNAs were selected due to their known relationships with apoptosis, inflammation and EMT [6,7,8,9,10,11], all topics related to PVR.

## 2. Results

Descriptions of participants are summarized in Table 1. During the follow-up period, four patients (17%) developed grade C posterior, type 1 (focal) PVR at the posterior pole. All PVR cases were initially detected at the 1-month follow-up visit.

LogMAR best-corrected visual acuity (BCVA) increased significantly from preoperative (median: 1.0 and IQR: 1.00–1.23) to postoperative, χ^2^(3) = 48.886, *p* < 0.001; in particular, the values significantly increased both at 1 month (median: 0.40 and IQR: 0.28–0.53; *p* < 0.001) and at 3 months (median: 0.20 and IQR: 0.18–0.33, *p* < 0.001), and remained unchanged at 6 months (median: 0.20, IQR: 0.18–0.33).

A significant difference was found in the expression of the 12 miRNAs analyzed (*p* < 0.001, η^2^_p_ = 0.853, ω^2^_p_ = 0.842, Cohen’s f = 2.311), as summarized in Table 2. In particular, the most expressed were miR-21, miR-let-7b and miR-210. Sex did not influence the expression of miRNAs (*p* = 0.227, η^2^_p_ = 0.090, ω^2^_p_ = 0.021, Cohen’s f = 0.146).

The 12 identified miRNAs were then tested for their hypothetical predictive value with all ophthalmological variables, with the following results (see Figure 1):none of the above 12 miRNAs significantly predicted the variable extent of macular detachment and status;miR-21 significantly predicted the time between the onset of symptoms and treatment (*p* = 0.017, r = 0.515), as did miR34 (*p* = 0.030, r = 0.624)—the more they were expressed, the greater the time;miR-21 significantly predicted the number of retinal breaks (*p* = 0.007, McFadden’s R^2^ = 0.298)—the more it was expressed, the lower the number of breaks;miR-26a correlated with postoperative BCVA at both 3 months and at 6 months (in both cases, *p* = 0.027, rho = 0.764)—in both cases, the more it was expressed, the higher the BCVA.

The expression of each of the 12 miRNAs was analyzed when comparing patients with and without severe myopia (Figure 2). The only significant difference was found for miR-183 (*p* = 0.012, Cohen’s d = 1.721, Hedges’ g = 1.641) and a tendency for let-7b (*p* = 0.104, Cohen’s d = 0.812, Hedges’ g = 0.703), whose expression was higher in those with severe myopia (ΔCt miR-183: 4.10 ± 1.16 vs. 6.11 ± 1.18; ΔCt let-7b: 1.13 ± 0.82 vs. 1.99 ± 1.25).

The expression of each of the 12 miRNAs was analyzed when comparing patients with and without PVR (Figure 2). No significant differences emerged, although trends were found for miR-146a, whose expression was higher in those who developed PVR (ΔCt: 3.31 ± 1.14 vs. 4.26 ± 0.69, *p* = 0.198, Cohen’s d = 1.007, Hedges’ g = 1.055).

## 3. Discussion

Under physiological conditions, active and passive forces promote normal adhesion between the neuroretina and the RPE. Rhegmatogenous retinal detachment is a condition in which fluid passes from the vitreous cavity into the subretinal space through a full-thickness retinal break, causing separation of the neuroretina from the underlying pigment epithelium. The most frequent pathological sequence in RRD involves vitreous liquefaction followed by posterior vitreous detachment, which in turn can cause a retinal tear at a site of tight vitreoretinal adhesion. We believe that the study of the SRF is important to understand the biological mechanisms triggered by retinal detachment, as the SRF composition might reflect the events occurring after the neuroretina separates from the pigment epithelial cells. Therefore, we focused our attention on a new class of gene expression regulators, miRNAs. The role of miRNAs as therapeutic targets or as disease markers is an active area of research. Indeed, in a recent pilot study, the expression of 754 miRNAs was evaluated in the vitreous humor of patients affected by PVR, identifying the altered expression of 20 miRNAs in the vitreous of patients with RD and different grades of PVR [11]. The authors identified 20 miRNAs with altered expression in one or more pathological groups; between these, they found let-7b-5p and miR-21-5p. The expression profiles of vitreous miRNAs in patients with PVR were compared to the profiles of those in unaffected individuals, demonstrating that different miRNAs can be used as molecular biomarkers for the disease.

In this study, for the first time, we showed the existence of possible correlations with PVR risk factors, as well as the existence of different expression profiles of miRNAs among patients with RRD with and without PVR. In fact, in the present study, we analyzed the expression profiles of a specific subset of miRNAs, reported in the literature as associated with the development of retinal fibrosis and angiogenesis [6,8,12,13,14]. Increased expression levels of miR-21 showed significant correlations with time from symptom onset to RRD surgery and with the number of retinal breaks. miR-21 has been shown to play a crucial role in a myriad of biological functions, including cardiovascular disease and inflammation. Lipopolysaccharide activates miR-21 expression in AEPR-19 cells and miR-21 in turn promotes the production of inflammatory cytokines such as MCP-1 and IL-6 [15]. Furthermore, miR-21 expression is often correlated with tumor progression in various cancer types [16] and fibrosis [17,18], and its pro-oncogenic and profibrotic role during disease progression has been reported. Conversely, down-regulation of miR-21 in AEPR-19 cells transfected with a miR-21 inhibitor promotes cell survival and reduces the incidence of apoptosis [19]. This could explain the relationship between PVR risk and retinal detachment duration. Moreover, miR-34, whose expression is correlated with the duration of detachment, is involved in the modulation of inflammation [20] and apoptosis [21]. In addition to miR-21, let-7b was among the most expressed miRNAs in our samples. The possible implication of the p53/let-7b/IGF-1 axis in the pathogenetic mechanisms underlying choroidal involution in subjects affected by ROP and other degenerative chorioretinopathies, such as geographic atrophy, has recently been described [22]. In subjects previously affected by ROP, choroidal thinning has been noted. Experimental models of oxygen-induced retinopathy were used to study this condition and its possible consequences, in which there was a reduction in pro-angiogenic factors including insulin-like growth factor-1 receptor (IGF1-R) with an increase in p53 expression. The expression of p53 is largely regulated by miRNAs and, in turn, p53 controls the expression of various miRNAs involved in apoptosis and angiogenesis. In particular, the down-regulation of IGF1-R is believed to be mediated at least in part by let-7b, and the reduction of p53 activity results in the reversal of senescence, normalization of IGF1-R expression and preservation of cell integrity [22]. Therefore, in light of the data revealed by our study, given the close correlation with p53 and the reciprocal regulation, we can hypothesize the involvement of let-7b in the mechanisms of cell cycle arrest, apoptosis and cellular senescence.

Another miRNA present in the subretinal fluid of our samples was miR-210. It is already known that miR-210 regulates many cellular processes, including apoptosis, cell growth and differentiation, cell cycle progression, DNA damage and mitochondrial metabolism. miR-210 regulates lipid metabolism and prevents photoreceptor neurodegeneration. In the Drosophila retina, the genetic deletion of miR-210 leads to the accumulation of lipid droplets and the degeneration of photoreceptors. These effects are associated with the abnormal activation of acetyl-coenzyme A synthetase (ACS), the target molecule of miR-210, while ACS reduction suppresses neurodegeneration defects. Together, these results reveal an unexpected role of miR-210 in controlling lipid metabolism and neuronal function [23].

Although no significant differences were found in miRNA expression in the analyzed samples from patients who did and did not develop PVR, a trend was observed concerning miR-146, which was less expressed in the samples of patients who had PVR. mir-146 is implicated in the regulation of innate and adaptive immune processes. Gene knockout studies have shown that miR-146a deficiency leads to excessive IL-6 and TNFα production. There is a wealth of laboratory evidence demonstrating that inflammation, particularly cytokines and inflammatory cells, may underlie the pathological changes leading to the development of PVR [24].

Finally, the significant expression of miR-183 in subretinal fluid from patients with high myopia is not surprising. Myopia predisposes patients to visual impairments such as myopic traction maculopathy and retinal detachment. Both genetic and environmental factors play a role in the development of myopia [25]. However, the exact etiology and mechanism of myopia are still unclear. Jacobi and Pusch [26] hypothesized that the majority of cases of myopia are probably not caused by defects in structural proteins, but by defects involving the control of structural protein expression. For this reason, we can hypothesize the existence of an association between miRNAs, important regulators of gene expression, and myopia, which needs further investigation. This study has some limitations to be highlighted. First, the small size of the sample might be justifiable by both the rare occurrence of RRD and the dramatic reduction in episcleral procedures in RRD management in the last few decades. Second, the search for miRNAs in the SRF made it impossible to obtain a control group of healthy subjects, since SRF collection is allowed only during scleral buckling procedures.

## 4. Materials and Methods

### 4.1. Patients and Sample Collection

A total of 24 eyes from 24 patients (17 males and 7 females; age 59 ± 11 years) who underwent scleral buckling procedures for rhegmatogenous retinal detachment were enrolled in the present prospective study. All diagnostic and surgical procedures were performed at the Ophthalmology Clinic of the University “G. d’Annunzio” of Chieti-Pescara between January 2017 and December 2019. This study was approved by the institutional ethics committee and patients provided signed informed consent for the use of their data. The study adhered to the principles of the Declaration of Helsinki. The exclusion criteria included systemic or ocular comorbidities that could affect the mechanisms underlying ocular fibrosis: diabetes mellitus, known rheumatic and autoimmune diseases, systemic treatments with corticosteroids or immunomodulatory drugs, the presence of any concomitant retinal disease, including age-related macular degeneration and vascular disease, previous ocular trauma, ocular tumors, any optic neuropathy including glaucoma, previous vitreoretinal surgery and any complicated cataract surgery.

Before surgery, all patients underwent a complete ophthalmological evaluation, including measurement of the best-corrected visual acuity using logarithmic ETDRS tables, tonometry, slit lamp biomicroscopy and indirect fundus ophthalmoscopy to assess the extent of detachment, location and number of breaks. In addition, all patients were tested by A- and B-scan ultrasound to calculate the axial length and to confirm the diagnosis and extent of RRD. Severe myopia was diagnosed when the axial length was longer than 26.5 mm.

Post-surgical visits were performed 1, 3 and 6 months after surgery. At each follow-up visit, patients underwent a full ophthalmological evaluation.

Twenty-four undiluted subretinal fluid samples were collected by evacuative puncture during the scleral buckling procedure, all performed by the same surgeon within three days of the first evaluation. During surgery, once the 360° peritomy was completed, the localization of the tears was determined by indirect ophthalmoscopy and the most appropriate drainage site was identified. Drainage was performed at the steepest point with respect to the extension of the detachment and sufficiently far from the retinal breaks and the localization of the vortex veins. A full-thickness sclerotomy was performed in order to drain subretinal fluid. Subsequently, a small incision was made through the choroid using a smooth punctal dilator. In order to facilitate SRF collection, drainage was performed via a 20-gauge cannula (Terumo SurFlash Polyurethane IV Catheter 20G; Tokyo, Japan) so that the fluid was collected immediately in a 3 mL syringe. At the end of the drainage procedure, a 6.0 Vicryl suture was placed across the two edges of the sclerotomy.

### 4.2. RNA Isolation

Approximately 250 μL of fluid was collected during surgery and immediately cooled on ice. Within 1 h, the subretinal fluid samples thus collected were centrifuged at 700 RCF for 10 min to remove cellular components, and the supernatant was resuspended in 700 μL of Qiazol Lysis Reagent and stored at −80 °C until further processing. The total RNA extraction from subretinal fluid was performed using the miRNeasy Micro Kit isolation kit (#217084, Qiagen, Hilden, Germany) following the manufacturer’s protocol. The total RNA concentration was quantified using the NanoDrop™. Quantification took place by measuring the absorbance at 2 different wavelengths (λ = 260 nm, indicative of the residual presence of nucleic acids; λ = 280 nm, indicative of the presence of proteins). Finally, the A260/A280 ratio was calculated, which was considered as an index of the quality of the extracted RNA.

### 4.3. MicroRNA Expression

Retro-transcription and quantitative real-time PCR (qRT-PCR) were carried out according to the Applied Biosystems TaqMan miRNA assay kit protocols (Life Technologies, Monza, Italy), as reported in [27]. The retro-transcription involved 20 ng of a “small” RNA, according to the Applied Biosystems high-capacity cDNA reverse transcription kit (#4368814). Then, the qRT-PCR for the miRNA expression levels was performed using the TaqMan probes and the specific TaqMan Universal Master Mix II, no UNG, in 96-well plates (#4440040, Applied Biosystems, Life Technologies, Monza, Italy), with an Applied Biosystems PRISM 7900 HT Sequence Detection System, in triplicate. miR-16 was used as the endogenous control because the expression levels were consistent across subretinal fluid samples, in the different samples. The mean ± S.D. of miR-16 expression levels was 23,301 ± 0,111. The specific miRNA sequence probes used (Applied Biosystems) were hsa-let-7b (#000378); has-miR-9 (#000583); has-miR-148a (#000470); has-miR-146a (#000468); hsa-miR-21 (#000397); hsa-miR-34a (#000426); hsa-miR-26a (#000405); hsa-miR-210 (#000512); hsa-miR-29b (#000413); hsa-miR-183 (#002269); hsa-miR-96 (#000186); hsa-miR-182 (#002334); hsa-miR-16–5p (#000391). The relative quantification of the miRNA targets was carried out using the ΔCt formula, according to the Ct method (Ct_miRNA of interest_ − Ct_miR-16_).

### 4.4. Statistics

Statistical comparisons were performed using Jamovi Version 1.2.27.0 software (https://www.jamovi.org). The assumptions were verified according to the Shapiro–Wilk test for the normality of the distributions, the observation of Q-Q graphs and the Levene test for homoskedasticity. For the comparison of Visus at different times, the Friedman test was used, with post-hoc comparisons according to the Durbin–Conover method. For the comparison between the 12 miRNAs, a general linear mixed model was used, setting participants as the random variable, with the residual maximum likelihood (REML) method, Satterthwaite method for degrees of freedom and Bonferroni correction for multiple comparisons. Then, we repeated the same analysis including sex as a factor. To test the predictive value of each of the miRNAs on the dependent variables of hours of detachment and time between onset of symptoms and treatment, linear regression methods were used; for the dependent variable of number of breaks, logistic regression on the ordinal variable was used; for the dependent variable of state of the macula, binomial logistic regression was used. Correlations between each of the 12 miRNAs and pre-operative and 1-, 3- and 6-month BCVA were computed according to the Spearman’s non-parametric method. Comparison of the expression of each of the 12 miRNAs in patients with and without severe myopia, and those with and without PVR, was performed according to Welch’s t-test for independent samples. For each comparison, the correct effect size was calculated according to literature indications [28,29], with the aid of an IT platform (https://effect-size-calculator.herokuapp.com (accessed on 25 January 2023)).

## 5. Conclusions

Within the context of new possibilities for understanding, diagnosing and treating retinal diseases, epigenetics has high potential [30]. MicroRNAs are a relatively new and powerful class of modulators that regulate gene expression, and their involvement in the pathogenesis of PVR is being studied and investigated. In this prospective study, we investigated the expression levels of specific miRNAs in the subretinal fluid of patients undergoing episcleral surgery for rhegmatogenous retinal detachment. It was possible to highlight that some miRNAs have significant correlations with known risk factors of PVR. Identifying the role of other microRNAs in EMT in RPE cells in vitro and in PVR in vivo could be helpful to understand the involvement of EMT-related gene expression more precisely. The extent of the detachment, long-lasting RDR, the presence of vitreous hemorrhage, large retinal tears and the presence of inflammation have been widely shown to be risk factors for the development of PVR. In particular, with the present study, we found the existence of a relationship between the duration of symptoms related to retinal detachment and miR-21 and miR-34, whose involvement in the regulation of cytokine expression is already described in the literature. Finally, we found that the expression of miR-146, a regulator of the immune response, tends to decrease in patients who develop PVR, confirming once again the role of miRNAs in the pathogenetic mechanism leading to the development of PVR.

## Figures and Tables

**Figure 1 ijms-24-03032-f001:**
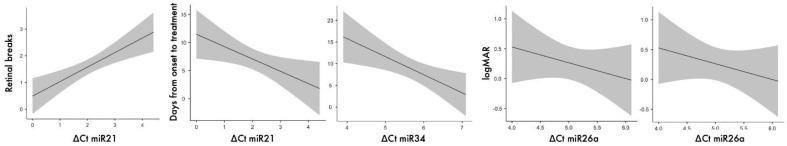
Main results from regression analyses; grey area around the linear tendency line is delimited by standard error.

**Figure 2 ijms-24-03032-f002:**
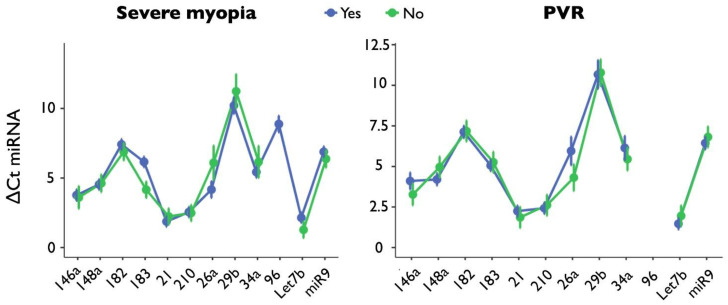
The graphs show miRNA expression stratified by severe myopia (on the **left**) or PVR (on the **right**); the 12 miRNAs are reported on the *x*-axis. Blue lines refer to the presence, while green lines indicate the absence, of myopia or PVR. For PVR, miR-96 values are missing because all the analyses in this group resulted in undetermined values.

**Table 1 ijms-24-03032-t001:** Characteristics of patients.

Patients, n = 24	
Age (years)	59 ± 11
Sex	
Male	71%
Female	29%
Side	
Right eye	65%
Left eye	35%
Retinal detachment size (No. of quadrants)	
≤2	65%
≥3	35%
Symptoms duration (weeks)	
≤1	62%
≥2	38%
Macular status	
On	21%
Off	79%
Retinal breaks (number)	
1	62%
2	21%
3	17%
BCVA, logMAR, median (IQR)	
Baseline	1.10 (1.00–1.23)
Severe myopia	21%

BCVA: best-corrected visual acuity (logarithm of the minimal angle of resolution); IQR: interquartile range.

**Table 2 ijms-24-03032-t002:** Expression of the 12 miRNAs in the analyzed SRF samples. Values are reported as mean (M) and standard deviation (SD). Grey columns indicate those miRNAs whose expression was greater compared to the others.

	ΔCt let7b	ΔCt miR9	ΔCt miR148a	ΔCt miR146a	ΔCt miR21	ΔCt miR34a	ΔCt miR26a	ΔCt miR210	ΔCt miR29b	ΔCt miR183	ΔCt miR96	ΔCt miR182
M	1.78	6.73	4.34	3.97	2.08	5.67	5.05	2.48	10.54	5.61	8.69	7.30
SD	1.21	1.59	0.93	0.92	0.99	0.80	0.57	1.81	1.35	1.45	1.56	1.03

## Data Availability

Not applicable.

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
