# Peer review of "MicroRNA Expression in Subretinal Fluid in Eyes Affected by Rhegmatogenous Retinal Detachment"

_ijms, 2023, doi:10.3390/ijms24033032_

Round 1
Reviewer 1 Report
The authors present an interesting prospective study comparing expression levels of 12 different miRNAs among subretinal fluid samples from 24 patients with RRD. The results were correlated with the pathogenesis of PVR and other eye exam findings. None of the miRNAs predicted the extent of macular detachment, miR-12 and miR-34 were correlated with increased time between onset of symptoms and treatment, miR-21 was correlated with a reduced number of retinal breakages, miR-26a was correlated with higher post-operative visus scores, miR-183 was correlated with severe myopia, while none of the miRNAs were correlated with the development of PVR (developed in 4 patients). This is an interested and informative study, is well-written, and indicates the potential importance of certain miRNAs in and around the retinas of patients with RRD.
Major Concerns:
1) This study focuses on 12 different miRNAs in the subretinal fluid, but it is not clear why the authors focus on these particular miRNAs. They do mention in the discussion section (line 128) that these were reported in the literature to be associated with retinal fibrosis and angiogenesis, but it is not clear why they decided on these 12 miRNAs. This should be clearly explained in the introduction before getting to the results.
2) Methods Line 244: Has miR-16 previously been tested as an effective endogenous control for comparing miRNA expression in subretinal fluid samples? If so, please provide a reference, or provide data showing that miR-16 expression levels are consistent across subretinal fluid samples. The reader needs to know why this is the best choice for a control.
3) Line 4 and Line 273: The authors seem to believe that miRNAs function in epigenetic pathways. Please consult the literature and ensure that there is a clear understanding of what epigenetic regulation is. Epigenetic means that there are heritable changes affecting gene expression that do not change the genetic sequence. miRNAs regulate molecular expression at the post-transcriptional level through interactions with messenger RNA, but these are not heritable changes that are passed from generation to generation. Epigenetics typically refers to DNA and histone modifications (methylation and/or acetylation).
Minor Concerns:
1) Lines 53-57: Should “DDR” and “DRR” be changed to “RRD”? These acronyms are very confusing.
2) Line 60 should begin with “Descriptions”.
3) Line 61: Remove the comma after “(focal)”.
4) Line 85: Please explain what “post-operative visus” means.
5) Figure 2: Please provide a longer caption that describes the figure and results so it is easy for the reader to interpret. Why are values for miR-96 missing? This should be explained in the caption and the main text.
Author Response
Please, see the attachment.

Reviewer 2 Report
The study is interesting and addresses the problem of rhegmatogenous retinal detachment from a novel perspective. However, there are certain aspects that should be improved.
The abstract does not include the most striking results of the study, nor the conclusion.
The methodology should explain why these 12 miRNAs were selected and not others.
In the RESULTS section, lines 71-73, the authors comment that "a significant difference was found in the expression of the 12 miRNAs analyzed...summarized in table 2". Which expression was compared to find significant differences? Does the preoperative and the postoperative? This must be clarified in the text and in the table. In addition, the table should show the expression of the 12 miRNAs at both times (preoperative / postoperative) and the p-value of each comparison.
It would have been nice to compare the expression according to sex.
The authors comment that it was impossible to use a control group of healthy subjects. Perhaps they could have used samples from donors.
There are some typos, so the authors should review the text of the manuscript. For example:
- lines 246-249: the name of the miRNAs is only correct in the case of let-7b (hsa-let-7b). In the rest of the miRNAs, "has" is written instead of "hsa".
Round 2
Reviewer 1 Report
The authors have appropriately addressed my concerns with the manuscript.
Reviewer 2 Report
The authors have adequately answered all the questions I raised in the previous review, sufficiently improving the manuscript.
Therefore, I recommend its publication.